# REM-related obstructive sleep apnea and vertigo: A retrospective case-control study

Po-Yueh Chen[1,2], Tzu-Ying Chen[3], Pin-Zhir Chao[2,3], Wen-Te Liu[2,4], Chyi-Huey Bai[5], Sheng-Teng Tsao[5], Yi-Chih Lin[2,3,5]*

**1** Department of Otolaryngology, Taipei Municipal Wan-Fang Hospital, Taipei Medical University, Taipei, Taiwan, **2** Sleep Centre, Shuang Ho Hospital, Taipei Medical University, New Taipei City, Taiwan, **3** Department of Otolaryngology, Shuang Ho Hospital, Taipei Medical University, New Taipei City, Taiwan, **4** Department of Chest, Shuang Ho Hospital, Taipei Medical University, New Taipei City, Taiwan, **5** Department of Public Health, College of Medicine, Taipei Medical University, Taipei, Taiwan

* 15677@s.tmu.edu.tw

**Data Availability Statement:** All relevant data are within the manuscript and S1 Dataset.

**Funding:** The authors received no specific funding for this work.

## Abstract

### Background

In recent population-based case-control studies, sleep apnea was significantly associated with a higher incidence (hazard ratio, 1.71) of vertigo and the risk of tinnitus was found to increase 1.36 times in patients with sleep apnea. The possibility that obstructive sleep apnea (OSA) might affect neurotological consequences was not noticed, until studies using polysomnography (PSG) for these patients.

### Objectives

The purpose of this study was to investigate the relationship between vertigo and OSA.

### Methods

The collected data among patients from May 1st, 2018 to October 31th, 2018 at Shuang Ho Hospital. Eligibility criteria included an age older than 20 years, a diagnosis of obstructive sleep apnea. The diagnosis of OSA was defined as an oxygen desaturation index of at least 5, was established with the use of polysomnographic examination at hospital. Patients were excluded from the study if they had head injury, brain tumour, headache history and hearing loss. Patients who had vertigo were labeled as Vertigo group. In the other hand, patients who had no dizziness were labeled as control group. 58 patients were in the Vertigo group, and 113 were in the control group.

### Results

After PSG examination, 58 patients who had vertigo, were diagnosed OSA (29 males, average age = 57.07 years old, BMI = 26.64, RDI = 24.69, ESS = 8.65), and 24 patients of them (41.3%) were REM-related OSA. Meanwhile, in the control group, 113 patients had OSA (92male, average age = 49.66 years old, BMI = 26.06, RDI = 35.19, ESS = 11.43), and 18 patients of them (15.9%) were REM-related OSA (Table 1). Therefore, patient who had vertigo, would have higher proportion of REM OSA (P<0.001).

**Competing interests:** NO authors have competing interests.

## Conclusions

The vertigo patients have a higher rate of REM-related OSA, and the acceptance rate to CPAP use is low. Further research is needed to explore novel therapeutic approaches, or combination of currently available non-CPAP therapies, in patients with REM OSA.

## Introduction

Vertigo is common among the general adult population [1]. A national telephonic survey in Germany revealed that the lifetime prevalence, 1-year prevalence, and incidence of vestibular vertigo were 7.8%, 5.2%, and 1.5%, respectively [2]. A national survey in Taiwan revealed that the prevalence of vertigo was 3.13 cases per 100 adults and that the prevalence and recurrence of vertigo increased significantly with age ($p < .001$) [3]. Vertigo can affect the ability to perform daily activities and negatively impact health-related quality of life [2, 4].

A recent population-based case–control study, 5,025 newly diagnosed with sleep apnea patients from the National Health Insurance Research Database of Taiwan, has been demonstrated sleep apnea is significantly associated with a higher incidence (hazard ratio, 1.71) of vertigo [5]. Vertigo symptoms of Ménière's disease patients with OSAS can be improved by CPAP therapy [6].

The possibility that obstructive sleep apnea (OSA) might have neurotological consequences was not noticed until patients were examined using polysomnography (PSG). OSA should be always suspected in elderly patients who show typical nighttime or daytime symptoms of sleep apnea. PSG is a necessary examination for the diagnosis of OSA in elderly patients [7].

OSA is a frequent disorder characterized by intermittent obstruction of the upper airway during sleep, occurring either during the rapid eye movement (REM) or nonrapid eye movement (NREM) stage. OSA may be predominantly a REM phenomenon in children and increasing in apnea severity may be secondary to changes in upper airway neuromotor control, changes in REM density [8], or upper airway muscle fatigue due to change in respiratory duty cycle [9]. During the REM stage, hypotonia of the upper airway muscles increases the risk for upper airway obstruction, and this is termed "REM-related OSA." Recently, interest has increased in the study of this subtype of OSA. The prevalence of REM-related OSA among patients with OSA undergoing PSG varies between 10% and 36% [10]. REM-related OSA may have worse cardiometabolic consequences than NREM-related OSA [4], and the current continuous positive airway pressure (CPAP) adherence guidelines may exclude most patients with REM-related OSA needing treatment.

To date, the association between vertigo and REM-related OSA remains unclear. In this study, we aimed to investigate the relationship between vertigo with sleep disturbance and REM-related OSA.

## Methods

### Study design

In total, 171 patients visiting the otology clinic and sleep clinic of Shuang Ho Hospital, Taiwan, were recruited from May 2018 through October 2018. Eligibility criteria included age more than 20 years and a diagnosis of OSA. According to the third edition of the International Classification of Sleep Disorders (ICSD-3), OSA defined as PSG-determined obstructive respiratory disturbance index (RDI) $\geq$ 5 events/h associated with the typical symptoms of OSA (e.g., unrefreshing sleep, daytime sleepiness, fatigue or insomnia, awakening with a gasping or

choking sensation, loud snoring, or witnessed apneas), or an obstructive RDI ≥ 15 events/h (even in the absence of symptoms). In addition to apneas and hypopneas index, the RDI includes respiratory effort-related arousals (RERAs) [11]. Patients with head injury, brain tumour, a history of headache, and hearing loss were excluded. We define vertigo group as patients who had two or more clinical visits due to vertigo within two years before the diagnosis of OSA. Besides, patients who experienced at least three months between two vertigo episodes. Fifty-eight patients were allocated to the vertigo group, and 113 patients who had no complaints of dizziness or vertigo within two years before the diagnosis of OSA were allocated to the control group.

Patient information collected consisted of age, gender, and body mass index (BMI). Daytime sleepiness was evaluated using the Epworth Sleepiness Scale (ESS). Data of other covariates, including respiratory disturbance index (RDI), NREM apnea–hypopnea index (AHI), REM AHI, apnea index, hypopnea index, sleep latency, N3 stage, REM stage, lowest oxygen saturation (LSAT), mean oxygen saturation (mean SAT), total sleep time with oxygen saturation below 90% (ST90), cumulative percentage of time spent at saturations below 90% (CT90), supine AHI, lateral AHI, mean heart rate (HR), arousal index, and periodic leg movement index (PLMI), were obtained through PSG. All of the PSG reports were scored by an experienced sleep technician and reviewed by a sleep medicine physician who was blinded to the study groups.

## Statistical analysis

Continuous data are presented as mean ± standard deviation (SD) and were analyzed using Student's $t$-test. Categorical data are presented as number (%) and were analyzed using the chi-square test. Fisher's exact test was used if the expected value was less than 5. Logistic regression analysis was performed to calculate the odds ratio (OR) and 95% confidence interval (CI). A $p$ value of $< .05$ was considered statistically significant. All the data were analyzed using SAS 9.4 (SAS Institute, Inc).

## Ethical approval

The retrospective study of data from patients who diagnosed obstructive sleep apnea was approved by the Institutional Review Board of Taipei Medical University (IRB Number: N202009009), which waived the requirement for informed consent of the patients involved because the study was retrospective and did not violate their rights or adversely affect their welfare.

## Results

Patient baseline characteristics of age, gender, BMI, and ESS score are presented in Table 1. The vertigo group had a greater number of female patients, older patients, and patients with lower ESS scores.

**Table 1. Baseline characteristics of study participants.**

| Characteristics | Total (N = 171) | Control group (N = 113) | Vertigo group (N = 58) | $p$ value |
|---|---|---|---|---|
| **Age (years)** | 52.18 ± 14.72 | 49.66 ± 14.25 | 57.07 ± 14.50 | .0017 |
| **Male patients, n/N (%)** | 120/171 (70.1) | 92/113 (81.4) | 29/58 (50.0) | .0001 |
| **BMI** | 26.26 ± 4.06 | 26.06 ± 4.02 | 26.64 ± 4.15 | .3781 |
| **ESS** | 10.54 ± 2.48 | 11.43 ± 1.89 | 8.65 ± 2.56 | < .0001 |

*BMI, body mass index; ESS, Epworth Sleepiness Scale.

*Comparison of the control group and the vertigo group was performed the t-test to evaluate the difference; except the gender were performed the chi square test.

**Table 2. Polysomnography results of participants.**

| Characteristics | Total (N = 171) | Control group (N = 113) | Vertigo group (N = 58) | p value |
|---|---|---|---|---|
| **RDI** | 31.63 ± 20.93 | 35.19 ± 21.00 | 24.69 ± 19.13 | .0003 |
| **REM AHI** | 43.10 ± 21.16 | 44.48 ± 21.60 | 40.50 ± 20.23 | .2496 |
| **NREM AHI** | 29.77 ± 22.15 | 33.88 ± 22.02 | 21.76 ± 20.29 | < .0001 |
| **REM-related OSA, n/N (%)** | 43/171 (25.1) | 18/113 (15.9) | 24/58 (41.3) | .0009 |
| **Apnea index** | 5.04 ± 7.58 | 6.02 ± 8.17 | 3.14 ± 5.89 | .0123 |
| **Hypopnea index** | 26.18 ± 18.18 | 28.63 ± 18.58 | 21.39 ± 16.50 | .0041 |
| **Sleep latency** | 25.13 ± 28.44 | 24.19 ± 28.19 | 26.97 ± 29.09 | .3567 |
| **N3 stage** | 3.81 ± 6.69 | 4.34 ± 6.93 | 2.78 ± 6.11 | .0785 |
| **REM stage** | 12.55 ± 6.48 | 12.17 ± 6.44 | 13.29 ± 6.54 | .4539 |
| **ODI** | 28.67 ± 21.34 | 30.96 ± 22.06 | 24.22 ± 19.28 | .0409 |
| **LSAT** | 82.41 ± 7.78 | 81.46 ± 8.07 | 84.26 ± 6.89 | .0191 |
| **Mean SAT** | 94.89 ± 6.84 | 94.72 ± 8.33 | 95.21 ± 1.76 | .1873 |
| **ST90** | 6.56 ± 17.08 | 7.94 ± 19.92 | 3.88 ± 8.88 | .0260 |
| **CT90** | 2.39 ± 5.70 | 2.81 ± 6.42 | 1.57 ± 3.85 | .0244 |
| **Supine AHI** | 37.26 ± 25.33 | 42.68 ± 25.10 | 26.51 ± 22.35 | < .0001 |
| **Lateral AHI** | 22.47 ± 23.92 | 27.72 ± 24.76 | 12.20 ± 18.46 | < .0001 |
| **Mean HR** | 65.31 ± 8.76 | 65.92 ± 9.39 | 64.12 ± 7.33 | .2043 |
| **Arousal index** | 21.43 ± 12.21 | 22.99 ± 12.01 | 18.40 ± 12.13 | .0020 |
| **PLMI** | 4.20 ± 11.28 | 3.48 ± 10.25 | 5.60 ± 13.03 | .6775 |

*RDI, Respiratory disturbance index; ODI, oxygen desaturation index; ST90, sleep time with SpO(2) < 90%; CT90, cumulative time percentage with SpO(2) < 90%.

LSAT, lowest oxygen saturation; mean SAT, mean oxygen saturation; PLMI, periodic leg movement index.

*Comparison of the control group and the vertigo group was performed the t-test to evaluate the difference.

PSG results for all the groups are presented in Table 2. The findings demonstrated that the vertigo group had significantly lower RDI, NREM AHI, apnea index, hypopnea index, ODI, ST90, CT90, supine AHI, lateral AHI, and arousal index values but higher REM-related OSA ($p = .0009$) and LSAT ($p = .0191$) values.

Logistic regression analysis was performed to calculate the OR to predict the risk for REM-related OSA. The ORs of potential risk factors for REM-related OSA are presented (Fig 1). If the patient with OSA was a woman, had symptoms of vertigo and tinnitus, and had an ESS score of less than 10, then the OR was increased to 3.3, 3.73, and 11.51, respectively. Moreover, if the patient with OSA had all of the aforementioned risk factors, the OR was increased to 35.00.

## Discussion

In this study, we found that patients with OSA and vertigo have a higher risk for REM-related OSA. Patients with sleep apnea appear to have an increased risk of vertigo [5]. The

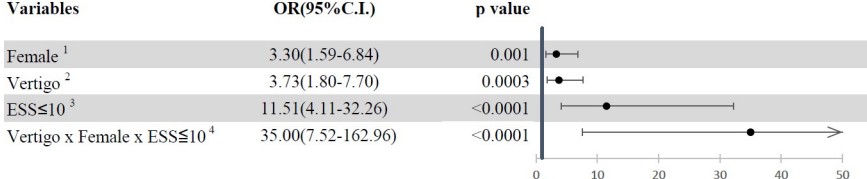

| Variables | OR(95%C.I.) | p value |
|---|---|---|
| Female [1] | 3.30(1.59-6.84) | 0.001 |
| Vertigo [2] | 3.73(1.80-7.70) | 0.0003 |
| ESS≦10 [3] | 11.51(4.11-32.26) | <0.0001 |
| Vertigo x Female x ESS≦10 [4] | 35.00(7.52-162.96) | <0.0001 |

**Fig 1. Odds ratios of potential risk factors of REM-related OSA.** 1: ref = male; 2: ref = non-vertigo; 3: ref = ESS>10; 4: ref = without any risk factors of female, vertigo or ESS≤ 10; OR, odds ratio.

accumulation of several factors may increase the severity of OSA during the REM sleep phase. During the REM stage, it would decrease the muscle tone in the upper airway, especially was genioglossus because of acetylcholine-mediated inhibition. This causes the muscles surrounding the upper airway to collapse easily. Moreover, the hypoxic and hypercapnic ventilatory drives decrease during REM sleep. The severity of REM-related OSA is usually reflected by events that are more frequent, of longer duration, and that are associated with greater oxygen desaturation. The pathophysiological mechanism of REM-related OSA remains unclarified. However, classifying OSA into non-stage specific and REM-related OSA is critical to prevent cardiovascular complications. Previous studies have demonstrated that OSA during REM sleep may increase the risk for cardiovascular complications compared with OSA during NREM sleep [12]. In the present study, the REM duration was less than 15% of the total sleep time, resulting in lower average values of RDI, ODI, ST90, CT90, and arousal index in the REM-related OSA group. This finding is in line with the result of an earlier study, which demonstrated that REM-related OSA was generally associated with lower arousal index values along with milder severity of OSA [10]. The reason for the vertigo group in our study demonstrating lower ESS scores could be the reduced severity of OSA in the NREM stage. This finding is in accordance with a previous study, which reported that an increase in NREM AHI, but not REM AHI, was associated with daytime sleepiness, as assessed by the multiple sleep latency test (MSLT) [13]. Additionally, even women reporting levels of daytime sleepiness similar to those reported by men are less likely to have an ESS score more than 10 [14]. Although it is unclear why these differences occur, it is possible that women have a different threshold for feeling sleepy or complain differently about sleepiness compared with men. The notion that REM-related OSA may be related to increased risks of cardiovascular, endocrine, and neurocognitive outcomes is based on at least 2 possibilities [15]. First, REM-related OSA may induce intermittently severe disease, and severe disease, even in limited doses, is adequate to increase the risks associated with adverse outcomes. Second, REM sleep constitutes the phase during which the brain and other end organs are particularly vulnerable; this can be attributed to the physiological features of the brain, including its precise neurochemical milieu, degree of neuronal synchrony, frequency of cortical local field potential oscillations, and autonomic tone, all of which aggregate to influence heart rate and rhythm and control blood flow to end organs.

Approximately 15% of patients with Meniere disease were diagnosed as having OSA [16]. Although OSA may significantly affect neurotological consequences, few studies have examined this topic. Improvement of sleep in patients with OSA is often associated with a decrease in vertigo-associated chronic dizziness. Studies have established that sleep (REM and slow-wave sleep) is essential to neuroplasticity processes, and this finding supports the premise of there being a correlation between the quality of sleep of patients following vestibular dysfunction and the effectiveness of vestibular compensation. An alteration in vestibular function can have repercussions on sleep quality, but the effect of such alteration on sleep remains unclarified. OSA, which induces sleep fragmentation, may lead to altered orexinergic activity. This may result in modulation of the vestibular nucleus activity, affecting postural instability and oculomotor control reported during the waking state [17].

In clinical practice, 4 hours of nightly CPAP use for 70% of nights is considered adequate compliance with the therapy. This translates into an average CPAP use of 2.8 hours every night. Reduced CPAP adherence and the predominantly untreated OSA during REM sleep may plausibly explain the negative effects of CPAP therapy on blood pressure control in randomized clinical trials. Critically, the use of CPAP for 3 or 4 hours from the time lights are turned off covers only 25% or 40% of REM sleep, respectively, and leaves most obstructive events during REM sleep untreated. Moreover, it is necessary to demonstrate that effective treatment of REM-related OSA, especially when associated with vertigo, leads to better patient

outcomes. One study showed REM related OSA is the risk factor for dropout from CPAP therapy, (odds ratio, 41.984) [18].

Clinically, polysomnography and home sleep apnea testing can be performed for the diagnosis of OSA. Watch-PAT in one kind of home sleep apnea testing, which can detect REM sleep well as PSG. As for pediatric patients, Watch-PAT is also an effective device for diagnosing sleep disordered breathing [19].

A previous Japanese study demonstrated that female sex and age more than 50 years were key risk factors for REM-related OSA [20]. These results indicate that hormonal changes in women might play a major role in REM-related OSA and might reflect its unknown pathophysiological characteristics. In recent population-based case–control studies, women had a significantly higher risk for vertigo than men with sleep apnea did [5]. Another study showed that women had lower NREM AHI compared to men, but had similar REM AHI as men. Besides it also found that the occurrence of REM-related OSA in women was 62% and in men was 24% [21].

A national survey in Taiwan demonstrated that vertigo was a frequent complaint in the general adult population and tended to recur, particularly among older women [3]. A clinical population study revealed that up to 40% of women with an AHI of more than 15/h had 1 or more of the classic OSA symptoms of snoring, witnessed apneas, and daytime sleepiness [22]. These findings are also apparent in our study.

This study has several limitations, which should be considered in future works. Other comorbidities, such as hypertension, diabetes, and rhinitis, were not considered when comparing the outcomes of patients with REM-related and NREM–related OSA. Moreover, most patients in the vertigo group were older and thus reported less daytime sleepiness, which could have resulted in potential bias.

## Conclusion

OSA patients with vertigo appear to have a higher rate of REM-related OSA, and the acceptance rate of CPAP treatment is low. Further research is needed to explore novel therapeutic approaches or the combination of currently available non-CPAP therapies, especially in patients with REM-related OSA. In addition, the odds ratio for REM-related OSA would be raised to 35(7.52–162.96) times, when OSA patients was a female, with vertigo symptom and less daytime sleepiness.

In the future, we plan to support these results and elucidate the vestibular–REM sleep interaction in a larger cohort, which could open avenues for further investigations.

## Supporting information

**S1 Dataset. The raw datasets and statistics for each analysis result graph and tables.** The raw data includes baseline characteristics of study participants and polysomnography results of participants.
(CSV)

## Acknowledgments

Lin YC conceived the idea; all authors have contributed to the design of the study; Chen PY, Chen TY, Lin YC, Chao PZ, Liu WT, Bai CH, and Tsao ST, collected and managed the data, including quality control. Tsao ST and Bai CH provided statistical advice on study design and analyzed the data. Lin YC supervised the conduct of the study. Chen PY and Lin YC drafted

the manuscript, and all authors contributed substantially to its revision. Lin YC takes responsibility for the paper as a whole.

## Author Contributions

**Conceptualization:** Pin-Zhir Chao, Yi-Chih Lin.

**Data curation:** Po-Yueh Chen, Tzu-Ying Chen, Pin-Zhir Chao, Yi-Chih Lin.

**Formal analysis:** Po-Yueh Chen, Chyi-Huey Bai, Sheng-Teng Tsao.

**Writing – original draft:** Wen-Te Liu, Yi-Chih Lin.

**Writing – review & editing:** Yi-Chih Lin.

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
