## [Decision Letter · Decision Letter 0]

11 Mar 2021

PONE-D-21-02581

Patients with Vertigo Have a Higher Risk of REM-Related Obstructive Sleep Apnea

PLOS ONE

Dear Dr. Lin,

Thank you for submitting your manuscript to PLOS ONE. After careful consideration, we feel that it has merit but does not fully meet PLOS ONE’s publication criteria as it currently stands. Therefore, we invite you to submit a revised version of the manuscript that addresses the points raised during the review process.

We look forward to receiving your revised manuscript.

Kind regards,

Giannicola Iannella, M.D

Academic Editor

PLOS ONE

Journal Requirements:

 "No"

3.We note that you have indicated that data from this study are available upon request. PLOS only allows data to be available upon request if there are legal or ethical restrictions on sharing data publicly. For information on unacceptable data access restrictions, please see http://journals.plos.org/plosone/s/data-availability#loc-unacceptable-data-access-restrictions.

4. Please amend your manuscript to include your abstract after the title page.

5.Thank you for submitting the above manuscript to PLOS ONE. During our internal evaluation of the manuscript, we found significant text overlap between your submission and the following previously published works:

https://link.springer.com/article/10.1007/s11325-018-1727-2?code=70a2e2ef-1617-4025-b1fc-e050d7cbf1c1&error=cookies_not_supported

https://www.sciencedirect.com/science/article/abs/pii/S1087079218300868?via%3Dihub

https://linkinghub.elsevier.com/retrieve/pii/S0012369217309303

https://www.hindawi.com/journals/crj/2018/9270329/

http://www.congre.co.jp/15jtcohns/program/abstract.pdf

Please revise the manuscript to rephrase the duplicated text, cite your sources, and provide details as to how the current manuscript advances on previous work. Please note that further consideration is dependent on the submission of a manuscript that addresses these concerns about the overlap in text with published work.

Additional Editor Comments:

Reviewers suggest revisions to your manuscript.

Best regards

Reviewers' comments:

Reviewer's Responses to Questions

**Comments to the Author**

1. Is the manuscript technically sound, and do the data support the conclusions?

Reviewer #1: Yes

Reviewer #2: No

2. Has the statistical analysis been performed appropriately and rigorously? 

Reviewer #1: Yes

Reviewer #2: Yes

3. Have the authors made all data underlying the findings in their manuscript fully available?

Reviewer #1: Yes

Reviewer #2: Yes

4. Is the manuscript presented in an intelligible fashion and written in standard English?

Reviewer #1: Yes

Reviewer #2: Yes

5. Review Comments to the Author

Reviewer #1: Maybe more fascinating title would be: REM-Related Obstructive Sleep Apnea and Vertigo: a Retrospective case-control study.

Introduction

- Add reference line 1 of vertigo prevalence.

- ‘’A national Taiwan study’’ insert p value data.

- ‘’ In recent population-based case–control studies’’Describe data of the population analyzed in the study to highilight the connection and add this reference: doi: 10.5664/jcsm.5080.

- ‘’ OSA is a frequent disorder characterized by’’ also cite other classical sleep stage in children and elderly patients using these references: doi:10.1164/ajrccm.162.2.9908058; DOI: 10.20452/pamw.15283 ; DOI: 10.1093/sleep/30.7.837

Study design

- Cite AASM guidelines for PSG describing better the procedures

Ethical approval is written in red: modify

Tables describe in wich group was performed the analysis: total vs control, total vs vertigo or what?

In the captations specify: Abbreviations: st90, sleep time…

Discussion PSG index is not anymore considered the only effective tool to evaluate sleep breathing disorder especially in children. Describe a different procedure and the data reported in this study doi: 10.1016/j.ijporl.2017.04.021.

Minor corrections:

Introduction

- correct ‘’were experiencing’’;

- modify center in centre;

- correct tumor in tumour;

- who were blindend in who was blinded;

- t test in t-test;

- or exinergic activity in orexinergic activity;

Discussion

- correct ‘’tends’’ in tended

Reviewer #2: The paper claims an innovative topic about sleep apnea. However, I have some doubts in relation to your evidences.

Introduction:

You cited your unpublished article without a sufficient definition of the parameters used to perform it.

Methods:

You said that 171 patients were enrolled with a diagnosis of OSA, but you did not classify the grade of vertigo disease of 58 patients and how it was diagnosed. Moreover, there were not given any information about the type of dizziness, the other symptomatology referred, and the previous eventual ear diseases.

Discussion

You asserted that “In our study, REM-related OSA had a CPAP acceptance rate of only 3.4%.”, but not described how you collected these data (duration of disease, duration of CPAP treatment etc…).

Could you clarify these topics?

6. PLOS authors have the option to publish the peer review history of their article (what does this mean?). If published, this will include your full peer review and any attached files.

Reviewer #1: No

Reviewer #2: No

---

## [Author Response · Author response to Decision Letter 0]

12 May 2021

We thank the reviewers for their constructive comments. We have revised the manuscript to address all the questions and comments raised by the three reviewers. We highlight changes made to the original version by setting the text colour to red. Our specific responses to each comment are as follows:

Responses to reviewers #1:

Maybe more fascinating title would be: REM-Related Obstructive Sleep Apnea and Vertigo: a Retrospective case-control study. 

We are grateful for all of your constructive comments, and we revised our title

Introduction

Add reference line 1 of vertigo prevalence. 

- Thank you for the comment. We add the reference in the article.

- [Please see line 65, introduction section; line 299, reference section]

‘’A national Taiwan study’’ insert p value data. 

- Thank you for the comment. We add p value in the article.

- [Please see line 70, introduction section]

‘’ In recent population-based case–control studies’’Describe data of the population analyzed in the study to highilight the connection and add this reference: doi: 10.5664/jcsm.5080.

- Thank you for the comment. We highlight the connection and add the reference in the article.

- [Please see line 72-76, introduction section; line 319-324, reference section]

- A recent population-based case–control study, 5,025 newly diagnosed with sleep apnea patients from the National Health Insurance Research Database of Taiwan, has been demonstrated sleep apnea is significantly associated with a higher incidence (hazard ratio, 1.71) of vertigo [5]. Vertigo symptoms of Ménière's disease patients with OSAS can be improved by CPAP therapy [6].

‘’ OSA is a frequent disorder characterized by’’ also cite other classical sleep stage in children and elderly patients using these references: 

DOI: 10.20452/pamw.15283 ; 

- Thank you for the comment. We add the reference in the article. 

[Please see line 79-81, introduction section; line 325-328, reference section]

- OSA should be always suspected in elderly patients who show typical nighttime or daytime symptoms of sleep apnea. PSG is a necessary examination for the diagnosis of OSA in elderly patients [7].

DOI: 10.1093/sleep/30.7.837 and doi:10.1164/ajrccm.162.2.9908058;

- Thank you for the comment. We add the reference in the article. 

[Please see line 84-87, introduction section; line 329-336, reference section]

- OSA may be predominantly a REM phenomenon in children and increasing in apnea severity may be secondary to changes in upper airway neuromotor control, changes in REM density [8], or upper airway muscle fatigue due to change in respiratory duty cycle [9].

Study design

Cite AASM guidelines for PSG describing better the procedures

- Thank you for the comment. We add the reference in the article. 

[Please see line 106-113,methods section; line 341-345, reference section]

- According to the third edition of the International Classification of Sleep Disorders (ICSD-3), OSA defined as PSG-determined obstructive respiratory disturbance index (RDI) ≥ 5 events/h associated with the typical symptoms of OSA (e.g., unrefreshing sleep, daytime sleepiness, fatigue or insomnia, awakening with a gasping or choking sensation, loud snoring, or witnessed apneas), or an obstructive RDI ≥ 15 events/h (even in the absence of symptoms). In addition to apneas and hypopneas index, the RDI includes respiratory effort-related arousals (RERAs) [11].

Ethical approval is written in red: 

- Thank you for the comment. We revised the color. 

[Please see line 141-145,methods section]

Tables describe in wich group was performed the analysis: total vs control, total vs vertigo or what?

- Thank you for the comment. We add the description below the tables 

[Please see line 153-154,result section; line 160-161, result section; table 1 and 2]

- *Comparison of the control group and the vertigo group was performed the t-test to evaluate the difference; except the gender were performed the chi square test.

In the captations specify: Abbreviations: st90, sleep time… 

- Thank you for the comment. We add the description below the tables 

[Please see line 156-159, result section; table 2]

- *RDI, Respiratory disturbance index; ODI, oxygen desaturation index; ST90, sleep time with SpO2< 90%; CT90, cumulative time percentage with SpO2 < 90%. LSAT, lowest oxygen saturation; mean SAT, mean oxygen saturation; PLMI, periodic leg movement index.

Discussion 

PSG index is not anymore considered the only effective tool to evaluate sleep breathing disorder especially in children. Describe a different procedure and the data reported in this study 

doi: 10.1016/j.ijporl.2017.04.021. 

- Thank you for the comment. We add the reference in the article.

[Please see line 239-242, discussion section; line 374-377, reference section ]

- Clinically, polysomnography and home sleep apnea testing can be performed for the diagnosis of OSA. Watch-PAT in one kind of home sleep apnea testing, which can detect REM sleep well as PSG. As for pediatric patients, Watch-PAT is also an effective device for diagnosing sleep disordered breathing [19].

Minor corrections:

Introduction

- correct ‘’were experiencing’’; � we delete this paragraph.

- modify center in centre; � we delete this paragraph.

- correct tumor in tumour; � [Please see line 48, abstract section; line 114, methods section]

- who were blindend in who was blinded; � [Please see line 131, methods section]

- t test in t-test; � [Please see line 135, methods section; line 153, results section]

- or exinergic activity in orexinergic activity; � [Please see line 224, discussion section]

Discussion

- correct ‘’tends’’ in tended� [Please see line 252, discussion section]

Responses to reviewers #2:

Introduction:

You cited your unpublished article without a sufficient definition of the parameters used to perform it.

- Thank you for the comment. We delete this paragraph. 

- [Please see line 91-95, introduction section]

Methods:

You said that 171 patients were enrolled with a diagnosis of OSA, but you did not classify the grade of vertigo disease of 58 patients and how it was diagnosed. Moreover, there were not given any information about the type of dizziness, the other symptomatology referred, and the previous eventual ear diseases.

- Thank you for the comment. We revised this paragraph. 

- [Please see line 114-121, methods section]

- We define vertigo group as patients who had two or more clinical visits due to vertigo within two years before the diagnosis of OSA. Besides, patients who experienced at least three months between two vertigo episodes. Fifty-eight patients were allocated to the vertigo group, and 113 patients who had no complaints of dizziness or vertigo within two years before the diagnosis of OSA were allocated to the control group.

Discussion

You asserted that “In our study, REM-related OSA had a CPAP acceptance rate of only 3.4%.”, but not described how you collected these data (duration of disease, duration of CPAP treatment etc…).

Could you clarify these topics?

- Thank you for the comment. We revised this paragraph. We delete “In our study, REM-related OSA had a CPAP acceptance rate of only 3.4%.” and change to “One study showed REM related OSA is the risk factor for dropout from CPAP therapy, (odds ratio, 41.984) [Hoshino T et al., 2018].”

[Please see line 235-237, discussion section]

- Hoshino T, Sasanabe R, Tanigawa T, Murotani K, Arimoto M, Ueda H, Shiomi T. Effect of rapid eye movement-related obstructive sleep apnea on adherence to continuous positive airway pressure. J Int Med Res. 2018 Jun;46(6):2238-2248. doi: 10.1177/0300060518758583. Epub 2018 Apr 3. PMID: 29614906; PMCID: PMC6023053.

---

## [Editor Report · Decision Letter 1]

24 May 2021

REM-Related Obstructive Sleep Apnea and Vertigo: a Retrospective case-control study.

PONE-D-21-02581R1

Dear Dr. Lin,

We’re pleased to inform you that your manuscript has been judged scientifically suitable for publication and will be formally accepted for publication once it meets all outstanding technical requirements.

Kind regards,

Giannicola Iannella, M.D

Academic Editor

PLOS ONE

Additional Editor Comments (optional):

The authors clarified many aspects suggested by the reviewers and improved the manuscript.

In my opinion it deserved the publication on PLOS-ONE
---

## [Editor Report · Acceptance letter]

3 Jun 2021

PONE-D-21-02581R1 

REM-Related Obstructive Sleep Apnea and Vertigo: a Retrospective case-control study. 

Dear Dr. Lin:

I'm pleased to inform you that your manuscript has been deemed suitable for publication in PLOS ONE. Congratulations! Your manuscript is now with our production department. 

Kind regards, 

on behalf of

Dr. Giannicola Iannella 

Academic Editor

PLOS ONE